# Miles Theory Revisited with Constant Vorticity in Water of Infinite Depth

## Christian Kharif * and Malek Abid

Aix-Marseille Université, CNRS, Centrale Marseille, IRPHE UMR 7342, F-13384 Marseille, France; abid@irphe.univ-mrs.fr
* Correspondence: kharif@irphe.univ-mrs.fr

**Abstract:** The generation of wind waves at the surface of a pre-existing underlying vertically sheared water flow of constant vorticity is considered. Emphasis is put on the role of the vorticity in water on wind-wave generation. The amplitude growth rate increases with the vorticity except for quite old waves. A limit to the wave energy growth is found in the case of negative vorticity, corresponding to the vanishing of the growth rate.

**Keywords:** shear instability; wind-wave generation; vorticity

## 1. Introduction

Wind-wave generation is a central problem in physical oceanography. Miles [1] is one the first to provide a theoretical framework to understand how wind energy is transferred to ocean surface waves. He considered the linear stability of an inviscid parallel shear flow described by a boundary layer in the air above a flat surface of water at rest. The transfer of energy occurs at the critical layer level where the wind velocity equals the phase velocity of the waves. Note that Hristov et al. [2] found from open ocean experiments that the structure of the wave-induced air flow is in agreement with the critical layer theory. Physically, this instability corresponds to a resonant interaction between the wave-induced pressure fluctuations and the surface waves. Valenzuela [3] and Kawai [4] introduced a wind-drift layer in water and solved the problem numerically by using the Orr–Sommerfeld equation in both air and water in the presence of surface tension instead of the Rayleigh equation. Beji and Nadaoka [5] investigated the effect of the shape of the wind profile on the wave growth rate and found appreciable differences. To avoid the problem of the critical layer singularity of the Rayleigh equation Stiassnie et al. [6] proposed a method of solution they called the exact approach. More recently, Young and Wolfe [7] considered the linear stability of an inviscid parallel shear flow in the presence of surface tension described by the double exponential profile solved analytically. In addition to the Miles instability in the air they found a second unstable mode called rippling mode which corresponds to an interaction between the surface waves and a critical layer in the water. Following Miles [1], we have considered a logarithmic wind profile in the air flow (modelling a turbulent wind) and a linear current profile in the water of arbitrary constant vorticity. Note that for a linear profile the rippling instability is not excited.

Our main goal is to revisit the Miles theory of wind wave generation at the surface of a pre-existing underlying water flow of constant vorticity in infinite depth. In Section 2.1 we compare the intrinsic phase velocities and energies of linear gravity waves in the presence of constant vorticity in both finite and infinite depth and come to the conclusion that when $kh > \pi$ the deviation between finite and infinite depths is weak. Section 2.2 is devoted to (i) the derivation of the Rayleigh equation (ii) the expression of the growth rate of the wave amplitude and the effect of negative vorticity of the water flow on the growth rate. Section 2.3 focuses on the effect of positive vorticity of the water flow

on the growth rate. We paid attention to the special case of the logarithmic wind profile in the air. A conclusion is made in Section 3.

## 2. Mathematical Formulation

The approach developed hereinafter is similar to those of Janssen [8] and Thomas [9] except that we now take the effect of a water flow in water of constant vorticity into account. For a detailed description of the method without water vorticity one can refer to Thomas [9].

*2.1. Preamble: Phase Velocity and Energy of Linear Gravity Waves at the Free Surface of a Shear Flow of Constant Vorticity*

The intrinsic phase velocity, $c_0$, of a linear gravity wave of wavenumber $k$, propagating in finite depth, $h$, at the free surface of a vertically sheared flow of constant intensity, $\Omega$, is

$$c_0 = -\frac{\Omega}{2k}\tanh(kh) + \sqrt{\frac{g}{k}\tanh(kh) + \frac{\Omega^2}{4k^2}\tanh^2(kh)}$$

where $g$ is the gravity.

Note that the vorticity is $-\Omega$.

In infinite depth the intrinsic phase velocity is

$$c_{0\infty} = -\frac{\Omega}{2k} + \sqrt{\frac{g}{k} + \frac{\Omega^2}{4k^2}}$$

There is no loss of generality if the study is restricted to waves propagating with positive phase velocities so long as both positive and negative values of $\Omega$ are considered.

Figure 1 shows the dimensionless intrinsic phase velocity deviation between finite and infinite depths as a function of the dispersive parameter $kh$ for several dimensionless values of $\Omega$. For $kh > \pi$ we can see that the deviation is weak. Generally, in laboratory experiments one considers that gravity waves of wavelength $\lambda < 2h$ behave like waves in infinite depth.

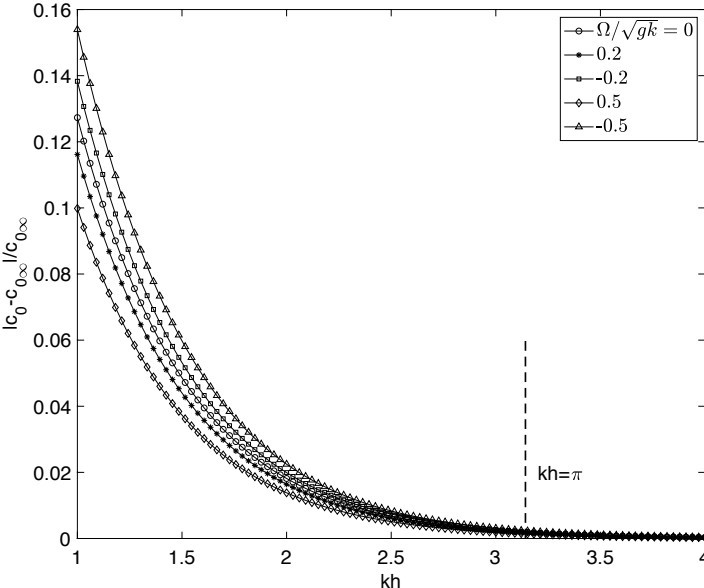

**Figure 1.** Dimensionless linear phase velocity deviation between finite and infinite depths for several values of the dimensionless vorticity.

Note that Teles da Silva and Peregrine [10] introduced a measure of water depth which influences the wave properties, $W_d = \tanh(kh)/(2k)$.

The condition for the presence of a critical layer in the water flow is $c_0 = \Omega z$ which only occurs for $\Omega < 0$. Teles da Silva and Peregrine [10] have shown that if there is a critical layer it is always at a depth $h_c > 2W_d$. Consequently, one can expect that the critical depth will not be in the layer of water which influences the wave if $kh_c > \tanh(kh)$.

The expressions of the excess of kinetic energy $T$ and potential energy $V$ are given by

$$2T = \int_0^{2\pi} \int_{-h}^{\eta} [(\phi_x + \Omega z)^2 + \phi_z^2] dz dx - \int_0^{2\pi} \int_{-h}^{0} (\Omega z)^2 dz dx$$

and

$$2V = \int_0^{2\pi} g\eta^2 dx$$

The of total energy is

$$E = T + V$$

In Figure 2 is shown the dimensionless excess of total energy deviation between finite and infinite depths of a linear gravity wave of wave steepness $ak = 0.10$ for several dimensionless values of $\Omega$. For $kh > \pi$ the deviation is weak.

Finally, linear gravity waves in finite depth propagating at the surface of a flow of constant vorticity behave nearly like waves in infinite depth if $kh > \pi$.

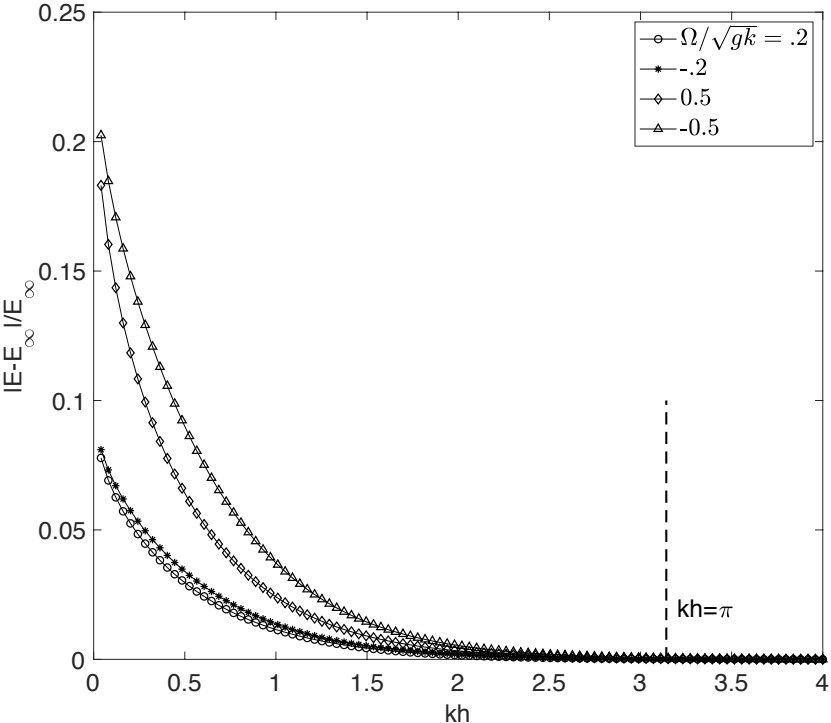

**Figure 2.** Dimensionless excess of energy deviation between finite and infinite depths for several values of the dimensionless vorticity of a linear wave of steepness $ak = 0.10$.

*2.2. Miles Theory in the Presence of Constant Vorticity in Water*

The inviscid governing equations of the flow in air and water are the following

$$\nabla \cdot \mathbf{u} = \mathbf{0} \tag{1}$$

$$\frac{d\mathbf{u}}{dt} = -\frac{\nabla p}{\rho} + \mathbf{g} \tag{2}$$

$$\frac{d\rho}{dt} = 0 \tag{3}$$

with

$$\frac{d}{dt} = \frac{\partial}{\partial t} + \mathbf{u} \cdot \nabla$$

and where $\mathbf{u}$ is the fluid velocity, $\rho$ is the fluid density, $p$ is the pressure and $\mathbf{g}$ is the acceleration due to gravity.

Equation (1) corresponds to mass conservation, Equation (2) is the Euler equation and Equation (3) means incompressible fluids.

We consider the linear stability of the following solution of the system of Equations (1)–(3) which corresponds to a flat air–water interface

$$\mathbf{u} = U_0 \mathbf{e}_x \tag{4}$$

$$\mathbf{g} = -g\mathbf{e}_z \tag{5}$$

$$\rho_0 = \rho(z) \tag{6}$$

$$p_0(z) = g \int \rho_0(z) dz \tag{7}$$

where $U_0$ corresponds to the velocity in the air and in the water, $\rho(z)$ corresponds to atmospheric density and water density and $\mathbf{e}_x$ and $\mathbf{e}_z$ are unit vectors in the $x$-direction and $z$-direction, respectively.

$$U_0(z) = \left\{ \begin{array}{lc} U_a(z) & , \quad z > 0 \\ U_w(z) & , \quad z < 0 \end{array} \right\}$$

where $U_a$ is the wind velocity and $U_w$ the flow velocity in the water.

$$\rho(z) = \left\{ \begin{array}{lc} \rho_a & , \quad z > 0 \\ \rho_w & , \quad z < 0 \end{array} \right\}$$

where $\rho_a$ and $\rho_w$ are the atmospheric density and water density, respectively.

Let us perturb the equilibrium given by Equations (4)–(7) with an infinitesimal perturbation[1]

$$U = U_0 + u' \tag{8}$$

$$p = p_0 + p' \tag{9}$$

$$\rho = \rho_0 + \rho' \tag{10}$$

---

[1]  The fluid medium is not homogeneous (air/water). Consequently, Equations (3) and (10) have been introduced to solve the problem.

Substituting the expressions (8)–(10) into Equations (1) and (2) and linearising gives

$$\frac{\partial u'}{\partial x} + \frac{\partial w'}{\partial z} = 0 \tag{11}$$

$$\frac{\partial u'}{\partial t} + U_0\frac{\partial u'}{\partial x} + w'U_0'(z) = -\frac{1}{\rho_0}\frac{\partial p'}{\partial x} \tag{12}$$

$$\frac{\partial w'}{\partial t} + U_0\frac{\partial w'}{\partial x} = -\frac{1}{\rho_0}\frac{\partial p'}{\partial z} + \frac{\rho'}{\rho_0^2}p_0'(z) \tag{13}$$

where $w'$ is the vertical component of the velocity perturbation.

The solutions of the linearized problem are sought in the following form (normal modes)

$$u' = u_1(z)\exp[i(kx - \omega t)] \tag{14}$$

$$w' = w_1(z)\exp[i(kx - \omega t)] \tag{15}$$

$$p' = p_1(z)\exp[i(kx - \omega t)] \tag{16}$$

$$\rho' = \rho_1(z)\exp[i(kx - \omega t)] \tag{17}$$

where $k$ and $\omega$ are the wavenumber and frequency of the perturbation, respectively.

Substituting the expressions (14)–(17) into the linearized equations gives the following Sturm–Liouville problem

$$\frac{d}{dz}\left(\rho_0 W^2\frac{d\psi}{dz}\right) - \left(k^2\rho_0 W^2 + g\frac{d\rho_0}{dz}\right)\psi = 0 \tag{18}$$

where $W = U_0 - c$, $c = \omega/k$ and $\psi = w_1/W$

The water flow is assumed to be vertically sheared with constant vorticity: $U_0 = U_s + \Omega z$, where the shear $\Omega$ and $U_s$ are constant. Without loss of generality we consider a frame of reference in which $U_s = 0$. Hence, Equation (18) reads

$$\frac{d}{dz}\left(\rho_w(\Omega z - c)^2\frac{d\psi_w}{dz}\right) - \left(k^2\rho_w(\Omega z - c)^2 + g\frac{d\rho_w}{dz}\right)\psi_w = 0$$

Let us assume $d\rho_w/dz = 0$, then

$$(\Omega z - c)\frac{d^2\psi_w}{dz^2} + 2\Omega\frac{d\psi_w}{dz} - k^2(\Omega z - c)\psi_w = 0 \tag{19}$$

First to avoid the existence of a critical layer in water, the study in deep water is restricted to waves with positive phase velocity and positive values of $\Omega$ (negative vorticity). Consequently, $\Omega z - c \neq 0$ and (19) reads

$$\frac{d^2\psi_w}{dz^2} + \frac{2\Omega}{\Omega z - c}\frac{d\psi_w}{dz} - k^2\psi_w = 0 \tag{20}$$

The case corresponding to $\Omega < 0$ (positive vorticity) will be discussed later in Section 2.3.

Equation (20) can be transformed to a reduced form

$$\frac{d^2\theta}{dz^2}(z) + q_1(z)\theta(z) = 0 \tag{21}$$

with the following change of variables

$$\psi_w(z) = \theta(z)\exp\left(-\frac{1}{2}\int_0^z \frac{2\Omega}{\Omega z' - c}dz'\right)$$

$$q_1(z) = -k^2 - \frac{1}{2}\frac{d}{dz}\left(\frac{2\Omega}{\Omega z - c}\right) - \frac{1}{4}\left(\frac{2\Omega}{\Omega z - c}\right)^2$$

that is

$$\psi_w = \frac{c}{c - \Omega z}\theta(z)$$

$$q_1(z) = -k^2$$

The reduced form of (20) is

$$\frac{d^2\theta}{dz^2}(z) - k^2\theta(z) = 0 \tag{22}$$

Hence the general solution of Equation (20) is

$$\psi_w(z) = \frac{c}{c - \Omega z}\Big(A\exp(kz) + B\exp(-kz)\Big)$$

The solution satisfying the condition $\lim\psi_w = 0$ as $z \to -\infty$ is

$$\psi_w(z) = \frac{c}{c - \Omega z}A\exp(kz))$$

Equation (18) is integrated between two points below ($z = 0-$) and above ($z = 0+$) the air–water interface

$$\rho W^2 \frac{d\psi}{dz}\Big|_{0-}^{0+} = \int_{0-}^{0+}\Big(k^2\rho_0 W^2 + g\frac{d\rho_0}{dz}\Big)\psi\, dz$$

with

$$\frac{d\rho_0}{dz} = (\rho_w - \rho_a)\delta(z)$$

where $\delta$ is the Dirac delta function.

$$\rho_a W^2(0+)\psi_a'(0+) - \rho_w W^2(0-)\psi_w'(0-) = g(\rho_a - \rho_w)\psi(0)$$

where $\psi(0) = \psi_a(0) = \psi_w(0)$ due to continuity of $\psi$.

$$\rho_a(U_a(0+) - c)^2\psi_a'(0+) - \rho_w(\Omega z - c)^2(0-)\psi_w'(0-) = g(\rho_a - \rho_w)\psi(0)$$

$$c^2(\rho_a\psi_a'(0+) - \rho_w\psi_w'(0-)) = g(\rho_a - \rho_w)\psi(0)$$

$$c^2\Big(\rho_a\psi_a'(0+) - \rho_w(Ak + \frac{A\Omega}{c})\Big) = g(\rho_a - \rho_w)\psi(0)$$

Because we consider linear waves, without loss of generality we can set $A = 1$

$$c^2\Big(\rho_a\psi_a'(0+) - \rho_w(k + \frac{\Omega}{c})\Big) = g(\rho_a - \rho_w)\psi(0) \tag{23}$$

The linear dispersion relation of gravity water waves on deep water in the presence of constant vorticity $kc^2 + \Omega c - g = 0$ is obtained from Equation (23) when wind effect is ignored.

Let $\epsilon = \rho_a/\rho_w$ and $c = c_0 + c_1\epsilon + \mathcal{O}(\epsilon^2)$ the Taylor series in $\epsilon$ in the presence of wind. Substituting the expansion of $c$ into Equation (23) gives

At $\epsilon^0$

$$kc_0^2 + \Omega c_0 - g = 0$$

At $\epsilon^1$

$$c_1 = \frac{c_0^2\psi_a'(0+) - g}{2kc_0 + \Omega}$$

Following Janssen [8] and Thomas [9], Equation (18), in the atmospheric medium, is reduced to the following form

$$\frac{d}{dz}\left(W_0 \frac{d\psi_a}{dz}\right) - k^2 W_0^2 \psi_a = 0 \tag{24}$$

$$\psi_a(0) = 1$$

$$\lim \psi_a(z) = 0 \quad \text{as} \quad z \to +\infty$$

where $W_0 = U_0 - c_0$.

The growth rate $\gamma_a$ of wave amplitude is

$$\gamma_a = \mathcal{I}m(kc_0 + kc_1\epsilon) = k\epsilon \mathcal{I}m(c_1)$$

$$\frac{\gamma_a}{\omega_0} = \frac{\epsilon c_0}{2kc_0 + \Omega} \mathcal{I}m(\psi_a'(0+))$$

where $\omega_0 = kc_0$ and $\mathcal{I}m$ denotes imaginary part.

Thomas [9] has shown that

$$\mathcal{I}m(\psi_a'(0+)) = \frac{i}{2} \mathcal{W}(\psi_a, \psi_a^*)(0+)$$

where $\mathcal{W}$ is the Wronskian given by

$$\mathcal{W}(\psi_a, \psi_a^*)(0+) = \psi_a(0+)\psi_a'^*(0+) - \psi_a'(0+)\psi_a^*(0+) = -2i\mathcal{I}m(\psi_a'(0+))$$

and $\psi_a^*$ denotes the complex conjugate.

Then

$$\frac{\gamma_a}{\omega_0} = i\frac{\epsilon c_0}{2(2kc_0 + \Omega)} \mathcal{W}(\psi_a, \psi_a^*)(0+) \tag{25}$$

Let $\chi = w/w(0)$ be the normalised vertical component of air velocity. Then, Equation (24) becomes the following Rayleigh equation

$$W_0\left(\frac{d^2}{dz^2} - k^2\right)\chi = W_0''\chi \tag{26}$$

$$\chi(0) = 1$$

$$\lim \chi(z) = 0 \quad \text{as} \quad z \to +\infty$$

The Rayleigh equation has a singular point where the phase velocity, $c_0$, of the waves equals the mean wind velocity $U_0$. Consequently, the height, $z_c$, of the critical layer in the atmosphere satisfies $U_0(z_c) = c_0$.

The growth rate can be rewritten as a function of the Wronskian of the solutions of the Rayleigh equation

$$\frac{\gamma_a}{\omega_0} = i\frac{\epsilon c_0}{2(2kc_0 + \Omega)} \mathcal{W}(\chi, \chi^*)(0+) \tag{27}$$

One can show that $\mathcal{W}'(\chi, \chi^*) = 0$. Consequently, the Wronskian is constant for $z > z_c$ and $z < z_c$ as well and may show a jump $\mathcal{W}(z_c + \varepsilon) - \mathcal{W}(z_c - \varepsilon)$, with $\varepsilon > 0$, at the critical height. Due to the boundary condition at infinity, $\lim \mathcal{W} = 0$ as $z \to +\infty$, $\mathcal{W}(z) = 0, \forall z > z_c$. Finaly, the jump is equals to $-\mathcal{W}(z_c - \varepsilon)$ and is given by the following expression

$$-\mathcal{W}(z_c - \varepsilon) = \lim I(\Delta, \varepsilon) \quad \text{as} \quad \Delta \to 0, \Delta > 0$$

with

$$I(\Delta, \varepsilon) = -4i \frac{W_{0c}''}{W_{0c}'} |\chi_c|^2 \arctan\left(\varepsilon \frac{W_{0c}'}{\Delta}\right)$$

The result is

$$-\mathcal{W}(z_c - \varepsilon) = -2i\pi \frac{W_{0c}''}{|W_{0c}'|} |\chi_c|^2$$

where $W_{0c}'' = W_0''(z = z_c)$, $W_{0c}' = W_0'(z = z_c)$ and $\chi_c = \chi(z = z_c)$.

The expression of the Wronskien is

$$\mathcal{W} = 2i\pi \frac{W_{0c}''}{|W_{0c}'|} |\chi_c|^2, \quad z < z_c$$

The normalised growth rate of surface wave amplitude is

$$\frac{\gamma_a}{\omega_0} = -\frac{\pi \epsilon c_0}{2kc_0 + \Omega} \frac{W_{0c}''}{|W_{0c}'|} |\chi_c|^2 \tag{28}$$

$$c_0 = -\frac{\Omega}{2k} + \sqrt{\frac{g}{k} + \frac{\Omega^2}{4k^2}}$$

Equation (28) can be written differently

$$\frac{\gamma_a}{\omega_0} = -\frac{\pi \epsilon c_0}{\sqrt{4gk + \Omega^2}} \frac{W_{0c}''}{|W_{0c}'|} |\chi_c|^2 \tag{29}$$

We assume the conservation of the momentum flux in the atmospheric boundary layer. Consequently, the wind profile is given by the folowing logarithmic law

$$U_a(z) = \frac{u_*}{\kappa} \ln(1 + \frac{z}{z_0}) \tag{30}$$

where $u_*$ is the friction velocity, $\kappa$ is the von Karman constant and $z_0$ is the roughness length of the air–water interface given by the Charnock relation $z_0 = \alpha_{ch} u_*^2 / g$.

Within the framework of a logarithmic law we obtain

$$\frac{W_{0c}''}{|W_{0c}'|} = -\frac{1}{z_0} \exp(-\kappa \frac{c_0}{u_*})$$

To derive the expression of the growth rate of the wave amplitude as a function of the wave age $c_0/u_*$ we use as reference velocity $u_*$ and reference length $u_*^2/g$. Let $c_* = c_0/u_*$, $k_* = u_*^2 k/g$, $\Omega_* = u_* \Omega/g$ and $z_{0*} = g z_0 / u_*^2$ be the dimensionless variables and parameters. Note that $z_{0*} = \alpha_{ch}$.

$$\frac{\gamma_a}{\omega_0} = \frac{\rho_a}{\rho_w} \frac{\pi}{z_{0*}} \frac{c_*}{2k_* c_* + \Omega_*} \exp(-\kappa c_*) |\chi_c|^2$$

Using

$$k_* = (1 - c_* \Omega_*)/c_*^2 \tag{31}$$

the dimensionless growth rate is rewritten as follows

$$\frac{\gamma_a}{\omega_0} = \frac{\rho_a}{\rho_w} \frac{\pi}{z_{0*}} \frac{c_*^2}{2 - c_* \Omega_*} \exp(-\kappa c_*) |\chi_c|^2 \tag{32}$$

where $c_*$ is the wave age and $-\Omega_*$ the dimensionless vorticity.

Note that $2 - c_* \Omega_* = 1 + k_* c_*^2 > 1$.

Using $z_{0*} = \alpha_{ch}$, Equation (32) reads

$$\frac{\gamma_a}{\omega_0} = \frac{\rho_a}{\rho_w} \frac{\pi}{\alpha_{ch}} \frac{c_*^2}{2 - c_* \Omega_*} \exp(-\kappa c_*)|\chi_c|^2 \tag{33}$$

The dimensionless amplitude growth rate depends only on the wave age and vorticity.

The Rayleigh Equation (26) is written in dimensionless form

$$(U_{a*} - c_*)(\frac{d^2}{dz_*^2} - k_*^2)\chi_* = U_{a*}'' \chi_* \tag{34}$$

where

$$U_{a*} = \frac{1}{\kappa} \ln(1 + \frac{z_*}{z_{0*}}) \qquad \chi_* = \frac{w_*}{w_*(0)} \qquad w_* = \frac{w}{u_*}$$

The dimensionless unknown $\chi_*$ is computed numerically by solving Equation (34) with the method of Conte and Miles [11]. The dimensionless growth rate of the wave amplitude $\gamma_a/\omega_0$ is calculated once the critical value of $\chi_*$ is known.

To check the validity of our approach we have compared our results in the absence of vorticity ($\Omega = 0$) with those of Beji and Nadaoka [5], Stiassnie et al. [6] and Kommen et al. [12]. Figure 3 shows the dimensionless growth rate, $\beta$, defined by Miles [1] as a function of the wave age

$$\beta = \frac{\rho_w}{\rho_a} \kappa^2 c_*^2 \frac{\gamma_E}{\omega_0}$$

where $\gamma_E = 2\gamma_a$ is the growth rate of the wave energy.

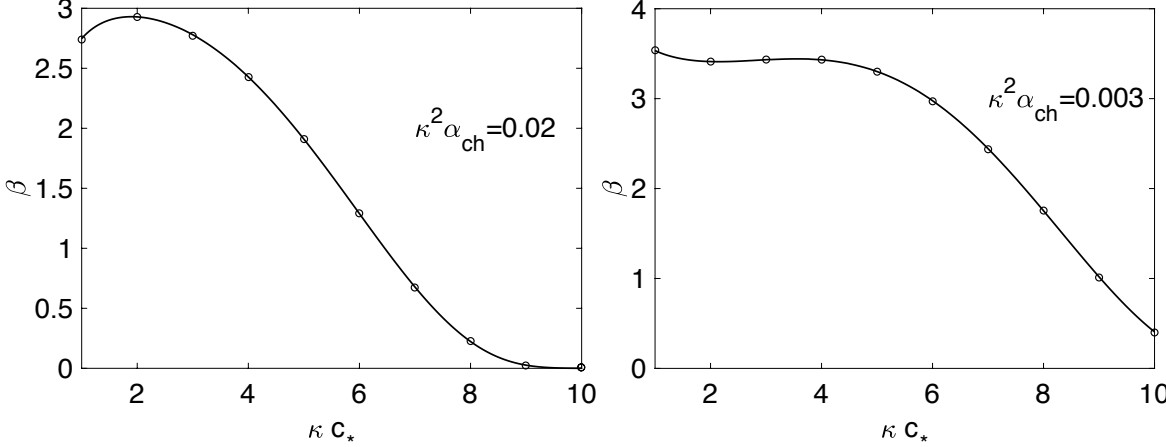

**Figure 3.** Dimensionless growth rate $\beta$ as a function of the wave age for two values of the Charnock constant without vorticity ($\Omega_* = 0$). Present results (solid line); Beji and Nadaoka [5] ($\circ$).

The agreement of our results with those of Beji and Nadaoka [5] who used a different method is excellent.

Figure 4 displays the dimensionless growth rate of wave energy as a function of the inverse of wave age in the absence of vorticity. The agreement of our results with those of Stiassnie et al. [6] is good whereas some deviation can be observed with those of Kommen et al. [12].

Figure 5 shows the dimensionless growth rate of the wave amplitude as a function of the wave age for different values of the dimensionless vorticity. We can see that the growth rate of waves generated at the surface of a vertically sheared flow of constant negative vorticity decreases as the intensity $\Omega_*$ increases and vanishes when a limit to the wave energy growth is reached. The wave age corresponding to $\gamma_a = 0$ can be determined easily as follows.

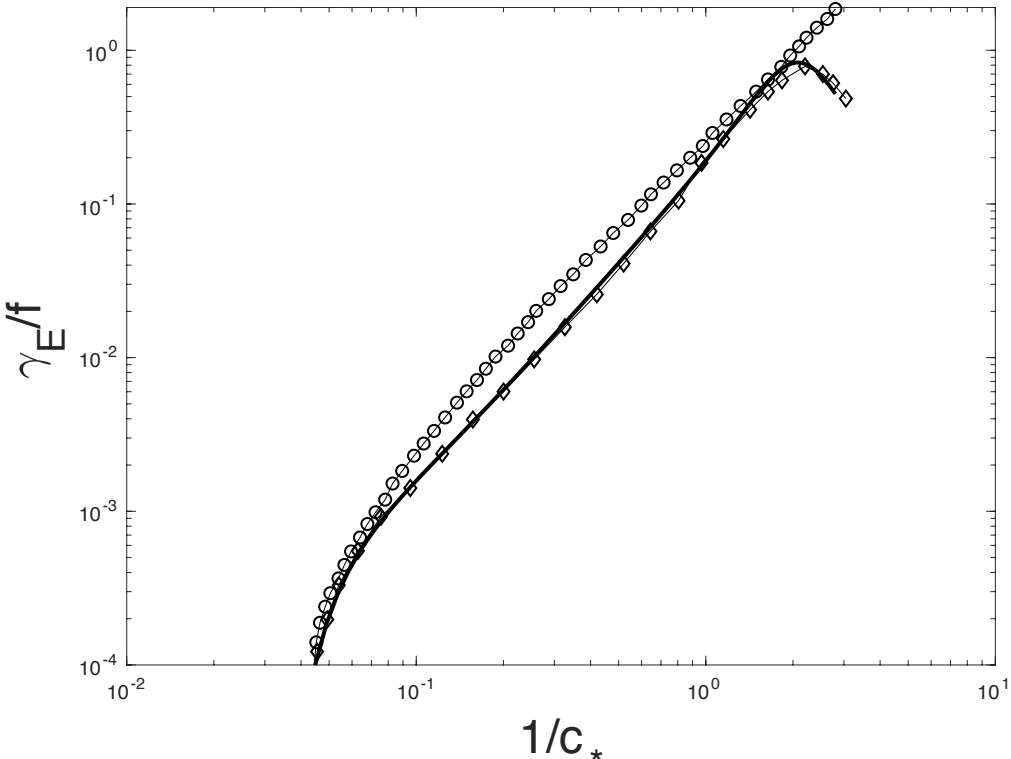

**Figure 4.** Dimensionless energy growth rate as a function of the inverse wave age without vorticity ($\Omega = 0$). Kommen et al. [12] ($\circ$); Stiassnie et al. [6] ($\diamond$); Present resuts (solid line).

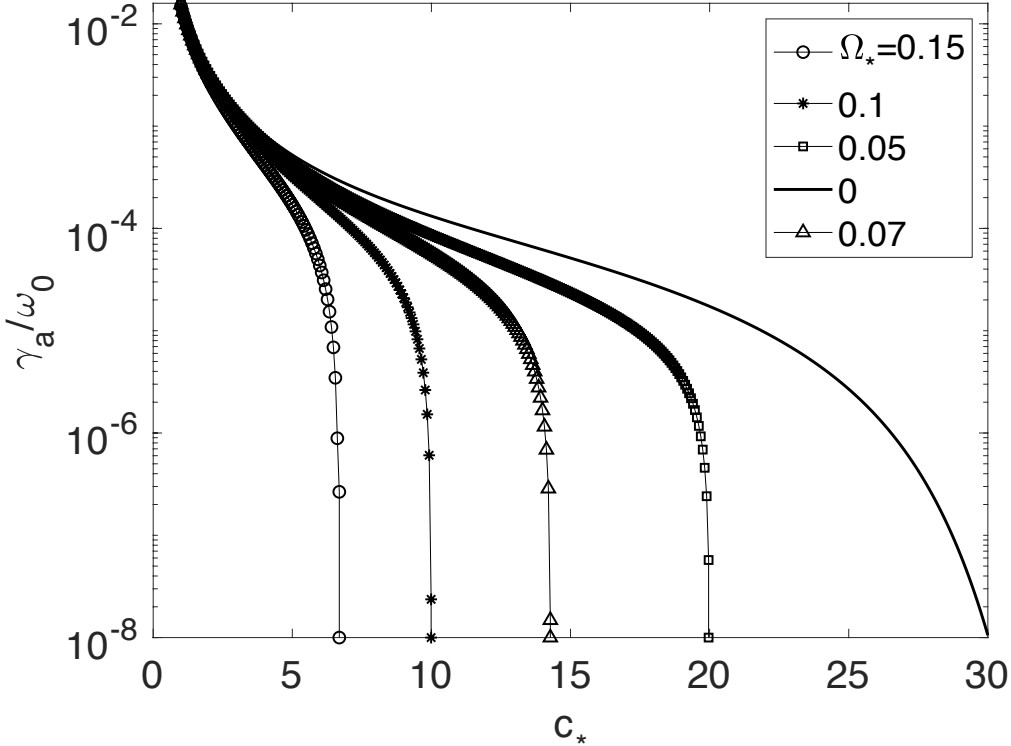

**Figure 5.** Dimensionless amplitude growth rate as a function of the wave age for several values of the dimensionless vorticity. The solid line corresponds to $\Omega_* = 0$.

The dispersion relation is

$$\omega_0^2 + \Omega\omega_0 - gk = 0$$

$$\lim \omega_0 = 0 \quad \text{or} \quad \lim \omega_0 = -\Omega \quad \text{as} \quad k \to 0$$

As emphasized above we consider wind waves with $\omega_0 > 0$ and $k > 0$. Consequently,

$$\lim c_0 = \lim \frac{\omega_0}{k} = \lim \frac{g}{\omega_0 + \Omega} = \frac{g}{\Omega} \quad \text{as} \quad \omega_0 \to 0$$

Hence

$$\lim c_* = \frac{1}{\Omega_*} \quad \text{as} \quad k_* \to 0 \tag{35}$$

Note that the limit wave age could be derived from Equation (31). This theoretical result suggests the existence of fetch limited wind wave growth in the presence of vertically sheared flows of constant vorticity. As the waves become increasingly longer, the growth rate decreases and vanishes.

The role of shear flows, in water of infinite depth, on the behaviour of wind waves is comparable to the role of finite depth without vorticity. Young and Verhagen [13] conducted field experiments showing depth limited wind wave growth. Later, Montalvo et al. [14] found numerically that finite depth limited growth is reached with wave growth rates going to zero (see Figure 1 of [14]).

### 2.3. Case of Positive Vorticity ($\Omega < 0$)

In the presence of positive vorticity ($\Omega < 0$) a critical layer occurs at $z_c = c_0/\Omega$. However, $U_w''(z) = 0$ cancels the logarithmic singularity at $z = z_c$ and consequently any instability mechanism in the water flow.

Figure 6 shows the dimensionless amplitude growth rate as a function of the wave age for different values of the vorticity. Three ranges of wave age are shown corresponding to $1 < c_* < 10$, $10 < c_* < 20$ and $20 < c_* < 30$. In the presence of a vertically sheared flow of constant positive vorticity the amplitude growth rate increases with the vorticity $-\Omega$, except for old waves.

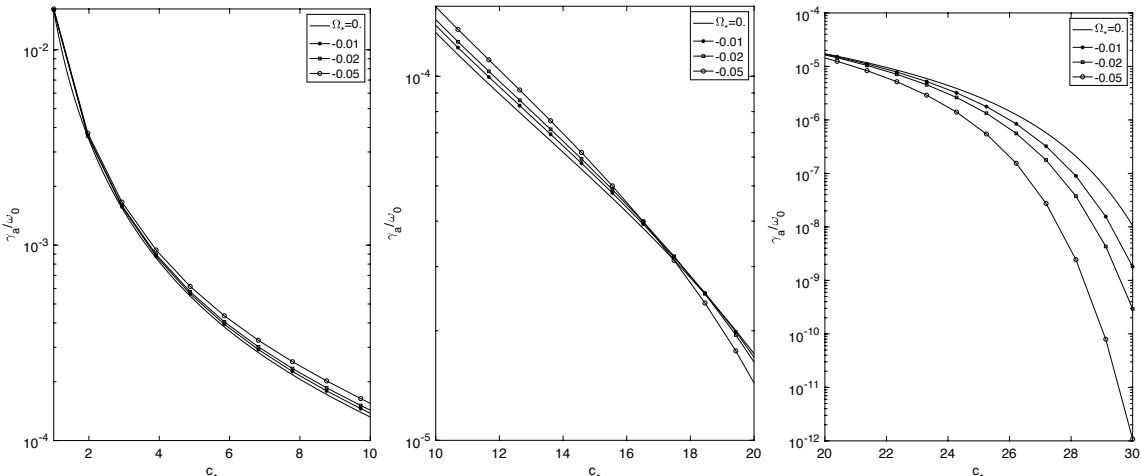

**Figure 6.** Dimensionless amplitude growth rate as a function of the wave age for several values of the vorticity.

## 3. Conclusions

We have revisited the Miles theory of wind wave generation by considering a pre-existing underlying water flow of constant vorticity in infinite depth. As $z \to -\infty$ the flow velocity becomes unbounded. Consequently, we have shown that gravity waves propagating at the surface of water flows of constant vorticity behave practically like waves in infinite depth providing that $kh > \pi$.

Beji and Nadaoka [5] and Stiassnie et al. [6] have investigated wind wave generation on a still water while we have considered the role of an undelying vortical water flow on wind wave generation. Valenzuela [3] and Young and Wolfe [7] investigated the growth rate of gravity-capillary waves in the presence of a depth varying negative vorticity of a wind drift water layer induced by the wind whereas we have considered wind wave generation at the surface of a pre-existing water flow of arbitrary constant water vorticity.

We found that the wave energy growth rate of wind waves, in the presence of negative (positive) water vorticity, decreases (increases) as vorticity decreases (increases). Notice that old waves behaves differently: their wave energy growth rates decrease as positive vorticity increases. Furthermore, in the case of negative vorticity the wave age limit corresponding to the vanishing of the wave energy growth rate depends on the value of the vorticity and is inversely proportional to the vorticity magnitude. These results emphasize the importance of the presence of vortical water flow on wind wave generation.

**Author Contributions:** Conceptualization, C.K.; Methodology, C.K.; Formal Analysis, C.K.; Writing original and revised draft, C.K.; Software, M.A.; Validation, M.A.; Investigation, M.A.; Visualization, M.A. All authors have read and agreed to the published version of the manuscript.

**Funding:** This research was funded by the Excellence Initiative of Aix-Marseille University—A*Midex, a French "Investissements d'Avenir programme" AMX-19-IET-010.

**Conflicts of Interest:** The authors declare no conflict of interest.

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
