# Peer review of "Miles Theory Revisited with Constant Vorticity in Water of Infinite Depth"

_jmse, doi:10.3390/jmse8080623_

Round 1

Reviewer 1 Report

This is a very interesting paper extending Miles classical theory to vorticity flows. The paper is well  written and the results are clearly presented. Previous work on the subject is summarised. The results are checked by recovering previous findings when vorticity is neglected. I recommend publication.

Author Response

We thank the referee for his positive comments on our paper.

Reviewer 2 Report

The manuscript in review presents a theoretical study on the influence of a Couette flow on the water side on the surface wave growth, while  also exploring the role of the finite depth.  For the wave perturbations the authors arrive to the Rayleigh equation and follow in the footsteps of Stiassnie et al. (2007) and Thomas (2012). The study concludes that for depths exceeding a half wavelength the waves propagate and grow as in deep water. Key findings on the growth rate are summarized in Figures 5 and 6. They indicate that the vorticity magnitude weakly affects the wave age range of large growth rates, i.e. the range of the growth rates above 1e-3, where most of energy transfer occurs. Only at low growth rates, i.e. below 1e-3, the curves fan out, i.e. the rates  notably diverge with the vorticity magnitude. The latter suggests that the effect vorticity magnitude on growth rates may not be easily observable in an experiment.   

The manuscript may require minor modifications. Both equations (1) and (3) express incompressibility, thus (3) appears redundant. So does the perturbation of fluid density (10). Repeated section title References seems redundant, too. On page 2 ‘scond unstable mode’ should probably be  ‘sEcond unstable mode’.

I recommend publication after minor revisions.

Reviewer 3 Report

Review on the manuscript entitled “Miles theory revisited with constant vorticity in water of infinite depth” written by Kharifa and Abida for the publication in Journal of Marine Science and Engineering (JMSE)

This paper presents novel and significant results that will be of great value to future investigators. I think it is important that this paper is published in JMSE, with minimal revisions, since it clearly fits to the journal with may provide a nice forum on generation of wind waves at the surface of a pre-existing underlying vertically sheared water flow of constant vorticity.

The authors present formulations on the role of the vorticity in water on wind-wave generation, revealing that the amplitude growth rate increases with the vorticity except for quite old waves. A limit to the wave energy growth is found in the case of negative vorticity (vanishing the growth rate). I recommend the acceptance of this manuscript as far as the authors revise the manuscript in a bit more attractable form, considering the suggestions below. Followings are general and specific comments on the manuscript.

Introduction: Recent work done by Young and Wolfe (2014; JFM) considered the linear stability of an inviscid parallel shear flow in the presence of surface tension as described in this manuscript. The Miles instability as well as the rippling mode was touched in that paper. Thus, it would be better for authors to describe how this study differs from that work in terms of revisiting the Miles theory in this section.

Conclusion: This work was concisely concluded in this section, describing new findings on the wave energy growth rate as an increasing function of the vorticity. However, there is no discussion emphasizing the significance of this work in comparison to several previous works – Young and Wolfe (2014; JFM), Beji and Nadaoka (2004; JFM), Valenzuela (1976; JFM), Stiassnie et al. (2007; JPO). It would be better to modify the title of this section to ‘Conclusion and discussion’.

Page 2, Line 8: ‘scond’ --> ‘second’

Page 4, Line 6: ‘ ~ potential energy are ~’ --> ‘ ~ potential energy V are ~’
